# Chemical and Biological Activity Profiling of *Hedyosmum* *strigosum* Todzia Essential Oil, an Aromatic Native Shrub from Southern Ecuador

**DOI:** 10.3390/plants11212832

**Published:** 2022-10-25

**Authors:** Luis Cartuche, James Calva, Eduardo Valarezo, Nayeli Chuchuca, Vladimir Morocho

**Affiliations:** Departamento de Química y Ciencias Exactas, Universidad Técnica Particular de Loja (UTPL), Calle M.Champagnat s/n, Loja 1101608, Ecuador

**Keywords:** enantiomeric distribution, chemical profiling, acetylcholinesterase, essential oil, thymol, *α*-thujene, *α*-pinene

## Abstract

The present study aimed to determine the chemical composition, enantiomeric distribution and the biological profile of *Hedyosmum strigosum* essential oil (EO). The antioxidant properties and anticholinesterase effect were measured by spectroscopic methods and antimicrobial potency assessed against 8 bacteria and two fungi. *H.* *strigosum* is a native shrub, particularly found in Ecuador and Colombia at 2000 to 3500 m a.s.l. Chemical composition was determined by GC-MS and GC-FID. A total of 44 compounds were detected, representing more than 92% of the EO composition. The main compounds were thymol (24.35, 22.48%), α-phellandrene (12.15, 13.93%), thymol acetate (6.59, 9.39%) and linalool (8.73, 5.82%), accounting for more than 51% of the EO. The enantioselective analysis revealed the presence of 5 pure enantiomers and 3 more as a racemic mixture. The EO exerted a strong antioxidant capacity, determined by ABTS assay, with a SC_50_ of 25.53 µg/mL and a strong and specific antimicrobial effect against *Campylobacter jejuni* with a MIC value of 125 µg/mL. A moderate acetylcholinesterase inhibitory effect was also observed with an IC_50_ value of 137.6 µg/mL. To the best of our knowledge this is the first report of the chemical composition and biological profile of *H*. *strigosum* EO.

## 1. Introduction

Chloranthaceae is the smallest family of vascular plants and it is the only family in the Chloranthales order. This family is composed mainly of lowering plants (angiosperms) and is considered one of the most primitive of its type [1]. The Chloranthaceae family consists of herbs, shrubs and trees, the species are characterized by the presence of secretory cells in the stems and leaves [2]. The family comprises 80 species belonging to the *Ascarina* (12 accepted species), *Chloranthus* (20 accepted species), *Hedyosmum* (44 accepted species) and *Sarcandra* genera (4 accepted species) [3]. Species of Chloranthaceae are naturally occurring in tropical and subtropical areas of Southeast Asia, Pacific Islands, New Zealand, Madagascar, Central and South America, and the West Indies [4]. In Ecuador, the Chloranthaceae family is represented only by the genus *Hedyosmum* [5].

Pharmacological investigations have revealed that the extracts of Chloranthaceae species are highly bioactive. Phytochemical studies showed the presence of terpenoids, flavonoids, coumarins, organic acids, amides, and sterols can be obtained from plants of this family [4]. Some species of the family Chloranthaceae have been used as folk medicine, especially in eastern Asia. A common use of these species includes: invigorating blood circulation and eliminating blood-stasis. The sarcandrae herb (*Sarcandra glabra*) is a folk medicine with a long history in China, widely used for its medicinal properties and it has been has been recorded in the Pharmacopoeia of the People’s Republic of China [6].

The genus *Hedyosmum* consists of 44 species of shrubs or trees, with branches hinged at the height of the nodes or bulging. Its leaves are opposite, petiolate, simple, toothed and with pinnate veins [7]. Most of the species of this genus are aromatic, for this reason, the genus name finds its origin in the Greek words, hedy (pleasant) and osmum (smell) [8]. This genus is represented mainly in America, and only one species is registered in Southeast Asia [9]. *Hedyosmum* species are widespread in low and high mountain rain forests at altitudes of 500 to 3000 m a.s.l. [10]. The species of this genus are distributed mainly in the mountains from the state of Veracruz (Mexico) to Brazil (through southern Mexico, Central America and the Andes of South America) [11,12,13].

In Ecuador, this genus is represented by at least 15 species and H. anisodorum, H. cuatrecazanum, H. cumbalense, H. goudotianum, H. luteynii, H. purpurascens, H. racemosum, H. scabrum, H. spectabile, H. sprucei, H. strigosum, and H. translucidum are mainly found in Andean forests and subparamos areas. H. purpurascens is the only endemic species occurring in the high Andean forest in the south of the country. All the reported species for Ecuador grown mainly in cloud forests between 600 and 3000 m a.s.l. [14,15].

Previous studies have reported non-volatile and volatile secondary metabolites (essential oil) in species of the genus *Hedyosmum*. Among the identified non-volatile compounds are flavonoids, hydroxycinnamic acid derivatives, neolignans, sesquiterpenes and sesterterpenes. The essential oils (EO) have been extracted from leaves, flowers, and fruits from plants of this genus [12,13,16]. Traditional uses assigned to *Hedyosmum* genus include the use as sources of firewood, construction materials and food (fruits). The leaves of some species are used in infusion as a medicinal preparations or aromatic beverages [7]. The pharmacological effects associated with species of this genus include analgesic, anxiolytic, antibacterial, anticancer, antidepressant, antinociceptive, antiplasmodial neuroprotective and sedative properties [17]. Some members of this genus have been reported to possess antispasmodic and digestive effects, and also, they were reported for the treatment of kidney problems and anxious stomach. In addition, folk medicine report traditional uses such as, headaches, snake bites, rheumatic joint pain, fever, and cold symptoms [2,18].

*Hedyosmum strigosum* Todzia (class: Equisetopsida C. Agardh; subclass: Magnoliidae Novák ex Takht.; superorder: Austrobaileyanae Doweld ex M.W. Chase & Reveal; order: Chloranthales R. Br.; family: Chloranthaceae R. Br. ex Sims; genus: *Hedyosmum* Sw.) is a native shrub or tree of Ecuador, also found in Colombia. In Ecuador, this species is distributed in the Andean and Amazon region between 2000 to 3500 m a.s.l., especially in the northern andean provinces of Cotopaxi, Imbabura, Napo, Pichincha and Tungurahua, and the eastern province of Sucumbíos [5]. The plant is popularly known as “Quinillo” or “Olloco,” and traditionally, the infusion of the leaves is drunk as an aromatic beverage, its fruits are food for birds, its branches and stems are used as fuel, and the stem is timber and is used for the construction of fences [18].

In south America, Ecuador is considered a megadiverse country because has many species per unit surface area. Currently, this country occupies the sixth position worldwide as a biodiversity hotspot [19]. However, there are few studies of its plant species, especially of the aromatic species of the *Hedyosmum* genus. The importance of the family and the genus, and the fact that only four species of this genus have been studied in Ecuador (*H. luteynii* [14], *H*. *racemosum* [2], *H*. *scabrum* [17] and *H*. *sprucei* [13]) encouraged us to determine a complete chemical profile of *H. strigosum* and validate its biological properties. To the best of our knowledge, this is the first report made for this specie, for that reason, the aim of this research was to determine the chemical composition, enantiomeric distribution, antimicrobial, antioxidant and anticholinesterase effect of the *H. strigosum* essential oil.

## 2. Results

The EO from *Hedyosmum strigosum* obtained by hydrodistillation exhibited a pale-yellow color and a yield of 0.66% (4.62 g from 700 g of plant material). The relative density was 0.9823 ± 0.004 g/mL, refractive index of 1.4893 ± 0.001 and a specific rotation of $\alpha_{D}^{20}$ of +21.21 ± 0.035 (*c* 10.25, *C*H_2_Cl_2_).

### 2.1. Chemical Composition

According to the GC-MS analysis, a total of 44 compounds were identified, representing 97.74% for DB5-MS and 92.27% for HP-Innowax, of the total oil composition. The main chemical identified compounds were thymol (24.35, 22.48%), α-phellandrene (12.15, 13.93%), thymol acetate (6.59, 9.39%), linalool (8.73, 5.82%) accounting for more than 51% of the chemical composition. Oxygenated monoterpenes were the main chemical groups with 46.96% and 43.33%, followed by hidrocarbonated monterpenes (31.13 and 28.52%) respectively (Table 1).

### 2.2. Enantiomeric Analysis

The results obtained for the enantiomeric analysis from *H. strigosum* essential oil are depicted in Table 2.

### 2.3. Antimicrobial Activity

According to the results displayed in Table 3, the most relevant outcome was obtained for *Campylobacter jejuni*, and the two dermatophytes with a MIC value of 125 µg/mL. The remaining microorganisms exhibited weak or null inhibitory effects at concentrations higher than 1000 µg/mL.

### 2.4. Antioxidant Capacity

The results obtained for the antioxidant capacity of the EO from H. strigosum are depicted in Table 4. Trolox was used as reference control with its corresponding IC_50_ value.

### 2.5. Anticholinesterase Activity

This is the first study of the inhibitory potency of H. strigosum, EO, against acetylcholinesterase, measured by the modified Ellman’s method. Results showed a moderate inhibitory effect with an IC_50_ value of 137.6 ± 1.02 µg/mL (Figure 1). Donepezil hydrochloride was used as positive control with an IC_50_ of 13.6 ± 1.02 µM.

## 3. Discussion

There are no previous reports about the extraction of EO from *H. strigosum*, however, a study from a related species, *H. luteynii*, from Chimborazo-Ecuador, reported a very low yield of the EO of ca. 0.07% by using the same extraction method [14]. In contrast, by means of microwave radiation-assisted hydrodistillation from *H. translucidum*, the EO yield obtained was 1.2% [20], approximately twice the yield obtained in our study.

According to Radice et al. [8] the main chemical compounds occurring in essential oils from representative species of the genus *Hedyosmum* (≥10%) corresponds to estragole (55.8%, leaves, *H.* scabrum), (*E,E*)-*α*-farnesene (32%, leaves, *H. costaricense*), germacrene D (32%, aerial parts, *H.* costaricense), *α*-pinene (24%, aerial parts, *H. angustifolium*), *β*-pinene ((23.5%, aerial parts, *H. angustifolium*), 1,8-cineole (20.5%, leaves, male plant, *H.* scabrum), linalool (16.5%, leaves, female plant, *H.* scabrum), sabinene (15.8%, leaves, male plant, *H, bomplandianum*), *β*-caryophyllene (15.5%, fresh aerial part, *H. sprucei*), pinocarvone (14.2%, leaves, male plant, *H. scabrum*), D-germacrene-4-ol (12.6%, leaves, male plant, *H. scabrum*), *δ*-3-Carene (12.1%, aerial part, *H.scabrum*), *α*-phellandrene (11.4%, leaves, *H. arborscens*), *α*-eudesmol (11.4%, leaves, *H, translucidum*), (*E*)-*β*-ocimene (10.8%, leaves, *H, bomplandianum*), bicyclogermacrene (10.6%, leaves, *H. arborescens*), *α*-bisabolene (10.3%, leaves, *H. bomplandianum*) and α-terpineol (10.2%, leaves, *H. brasiliense*).

There are no reports about the antimicrobial activity of *H. strigosum*, however, *H. brasiliense,* a related species, showed a good profile of inhibition against several pathogenic bacteria such as *Bacillus subtillis*, *Staphylococcus aureus*, *Staphylococcus saprofiticus* and dermatophytic fungi such as *Microsporus canis*, *Trichophyton rubrum* and *Trichophyton mentagrophytes* with MIC values ranging from 0.31 to 0.12% *v/v* (3.12 to 1.25 mg/mL ca.) [9]. There are no extended criteria to value the potency of MIC values, however, Van Vuuren and Holl [21] suggested a detailed classification for extracts and essential oils and found that a MIC value between 101 to 500 µg/mL can be considered as a strong activity and according to our report, the activity for *C. jejuni* and the two dermatophytes lies between this range.

The *H*. *strigosum* EO as shown in Table 4, reveals a strong scavenging effect for ABTS radical, meanwhile a weak or null scavenging capacity for DPPH radical. This can be due to the mechanistic differences of both assays. ABTS cation radical is stabilized by electron transfer mechanism from antioxidants present in samples and for DPPH assays, this radical can be mainly stabilized by hydrogen atom transfer mechanism [22]. The occurrence of the aliphatic monoterpene α-phellandrene in concentrations higher to 28% in the species *H. racemosum*, another Ecuadorian related species, and its null antioxidant activity [2] could suggest at least in our research that, the antioxidant capacity could be related to the main compound, thymol, which demonstrated a strong antioxidant effect as described by Torres-Martínez et al. [23] against ABTS assay with a scavenging effect higher than 96% at a 0.1 mg/mL dose. However, according to Beena et al., thymol exerted a moderate antioxidant effect in the DPPH assay with a SC_50_ value of 167.57 µg/mL [24] which can agree with our results.

Enantiomeric analysis showed the presence of 5 pure enantiomers and 3 pairs found as a racemic mixture, other equally interesting studies showed enantiomeric compositions of EO from the *Hedyosmun* genus, such as; *H. scabrum* (Ruiz & Pavon) leaves collected in Ecuador and Bolivia, and *H. angustifolium* collected in Bolivia the presence of similar enantiomers such as β-pinene, sabinene, limonene and linalool [17,25].

Regarding to the anticholinesterase effect, the *H. strigosum* EO exerted a moderate inhibitory potency with an IC_50_ value of 137.6 µg/mL, much higher than the reported effect of *H*. *barisiliense* EO which exhibited an inhibition percentage of 69.82% at a dose of 1 mg/mL. Many species of *Hedyosmum* genus have been explored for their antibacterial or antioxidant potential and their chemical profile but little or nothing is said about their inhibitory potential against acetylcholinesterase, however, information related to chemical entities isolated from the EO of this genus and their inhibitory potential can be found in literature, such as thymol which is the characteristic occurring compound in *Thymus vulgaris* but, it was found as the majority compound in our EO. As reported by Jukic et al. [26], thymol presented an IC_50_ of 0.74 mg/mL and linalool was inactive at a dose higher than 1 mg/mL, surprisingly, carvacrol, the structural isomer of thymol, exerted a ten times higher potency with an IC_50_ of 0.063 mg/mL against acetylcholinesterase, suggesting that the hydroxyl functionality in -orto position, like occurring in carvacrol, plays an important role for the AChE inhibitory effect.

Despite its low concentration in the EO of *H. strigosum*, terpenes like terpinolene, terpinen-4-ol and *E*-caryophyllene (< to 1%) can be collaborating for the moderate AChE inhibitory effect observed. As reported by Bonesi et al., terpinolene presented an IC_50_ of 156.4 µg/mL [27], meanwhile, terpinen-4-ol and *E-caryophyllene* exhibited a moderate effect with inhibition percentages of 21.4% at 1.2 mM and 32% at 0.06 mM doses, respectively. There is no report about the AchE inhibitory effect of α-phellandrene, the second major compound occurring for *H. strigosum* EO, however, in the same study of Bonesi, the AChE inhibitory effect of β-phelandrene is reported with an IC_50_ value of 120.2 µg/mL. highlighting the importance of terpenes containing EO and their AChE inhibitory capacity.

## 4. Materials and Methods

### 4.1. Plant Material

The aerial parts of *Hedyosmum strigosum* were collecteced in Villonaco hill, Loja, province at 2720 m a.s.l. at 3°59′42.62” S and 79°16′02.60” W coordinates. The identified specimen was located at the herbarium of the Universidad Técnica Particular de Loja with voucher code HUTPL14299. The aerial parts were cleaned and dried at 34 °C for 6 days until the plant material was used for distillation.

### 4.2. Essential Oil Distillation

The aerial plant material was submitted to hydrodistillation in a Clevenger-type apparatus for a period of 3 h. Anhydrous sodium sulfate was added to the flask containing the EO with moisture to remove it and then, the EO properly labeled was stored at −4 °C until was used in the biological and chemical assays.

### 4.3. Physical Properties Determination

Three physical properties were determined for the EO. Relative density (RD_20_), according to the AFNOR NFT75-111 standard, with an 1 mL pycnometer and an analytical balance (Mettler AC100 model), refractive index (RD_20_), according to the ANFOR NF 75-112 25 standard in an ABBE refractometer (Boeco, Germany) and specific rotation $[\propto_{514}^{20}],\;$according to the ISO 592-1998 standard in a Hanon P 810 automatic polarimeter, were carried out. All measurements were performed by triplicate.

### 4.4. Chemical Profiling

#### 4.4.1. GC-MS

Chemical analysis of *H*. *strigosum* EO was carried out using a gas chromatography-mass spectrometry (GC-MS) model Trace1310 gas chromatograph, coupled to a simple quadrupole mass spectrometry detector model ISQ 7000 (Thermo Fisher Scientific, Walthan, MA, USA). The mass spectrometer was operated in SCAN mode (scan range 35–350 m/z), with the electron ionization (EI)source 70 eV.

A non-polar column, based on 5%-phenyl-methylpolysiloxane, DB-5 ms (30 m long, 0.25 mm internal diameter, and 0.25 μm film thickness), was used for the qualitative and quantitative analysis of EO. The thermal program was as follow: initial temperature 60 °C for 5 min, followed by a gradient of 2 °C/min until 100 °C, then at 3 °C/min until 150 °C, and at 5 °C/min until 200 °C. Finally, a new gradient of 15 °C/min until 250 °C was applied. The final temperature was maintained for 5 min. GC purity grade helium 5.0 grade ultra-pure, was used as carrier gas, set at the constant flow of 1 mL/min [28].

#### 4.4.2. GC-FID

Quantitative analysis was performed on an Agilent Gas Chromatograph (6890 series) with a flame ionization detector (CG-FID) the chromatographic conditions were the same as in the GC-MS analysis. The average percentage of each component EO was calculated from the area of the corresponding CG-FID peak with respect to the total area of peaks without applying any correction factor. The values and standard deviations were calculated from the results of the three injections [29].

#### 4.4.3. Enantioselective Analysis

Enantioselective analysis was carried out through an enantioselective from Mega, MI, Italy capillary column, based on 2,3-diethyl-6-tert-butyldimethylsilyl-β-cyclodextrin (25 m × 250 μm internal diameter × 0.25 μm phase thickness). GC method was as follows: initial temperature was 60 °C for 2 min, followed at 2 °C/min until 220 °C, that was maintained for 2 min. The homologous series of n-alkanes (C9–C25) was also injected, in order to calculate the linear retention indices. The enantiomers were identified determined by injection of enantiomerically pure standards [30].

### 4.5. Antimicrobial Activity

The antimicrobial activity and Minimal inhibitory concentration values were determined by means of the broth microdilution technique with four gram positive bacteria (*Enterococcus faecalis* ATCC^®^ 19433, *Enterococcus faecium* ATCC^®^ 27270, *Staphylococcus aureus* ATCC^®^ 25923, *Listeria monocytogenes* ATCC^®^ 19115), four gram negative bateria (*Escherichia coli* (O157:H7) ATCC^®^ 43888, *Pseudomonas aeruginosa* ATCC^®^ 10145, *Salmonella enterica* serovar *Thypimurium* ATCC^®^ 14028) and two fungi (*Trichophyton rubrum* ATCC^®^ 28188, *Trichophyton interdigitale* ATCC^®^ 9533) as microbial assay models. Two-fold serial dilution method was used to achieve concentrations ranging from 4000 to 31,25 µg/mL and a final inoculum concentration of 5 × 10^5^ cfu/mL for bacteria, 2.5 × 10^5^ cfu/mL for yeast and 5 × 10^4^ spores/mL for sporulated fungi were used. Meller Hinton II (MH II) for bacteria and Sabouraud broth for fungi as assay media were used. The method followed was previously described for our research group. The culture of *Campylobacter jejuni*, was activated by adding horse serum at 5% in Tioglycolate medium and incubated at anaerobic conditions provided by a Campygen sachet (Thermo, scientific) at 37 °C for 48 h. The broth microdilution test was performed in MH II supplemented with 5% horse serum (Thermo) at the same conditions previously described as oil sample dilution and bacterial inoculum, and anaerobic conditions as described fully by our research group [31].

Antimicrobial commercial agents were used as positive control as follows: Ampicilin 1 mg/mL solution for *S.aureus*, *E. faecalis* and *E. faecium*. Ciprofloxacin 1 mg/mL solution for *P.a*eruginosa, *S. enterica*, *E. coli*. and *L. monocytogenes*, finally, Itraconazol 1 mg/mL for the two fungi.

### 4.6. Antioxidant Capacity

#### 4.6.1. The 2,2-Diphenyl-1-Picril Hydrazyl Radical Scavenging Assay

The procedure was adapted from the methodology suggested by Thaipong et al. [32] and fully developed for our research group in previous reported jobs [31]. Briefly, a stable solution of DPPH in MeOH was prepared at a concentration of 625 µM. Working solution was prepared by adjusting an aliquot of the stable solution in methanol until an absorbance of 1.1 ± 0.02 was reached. An EPOCH microplate reader was used for absorbance reading at 515 nm and 96 microplates were used for the test. 270 µL of the working solution were allowed to react with 30 µL of the sample solution for 1 h at room temperature in darkness. The same procedure was followed for the blanks and positive control (Trolox). A solution representing 100% of the DPPH radical was prepared with 270 µL of working solution and 30 µL of MeOH. Sample solutions of essential oil were prepared by dissolving 10 mg of EO in MeOH. Three more 10 × dilutions were included to calculate the SC50 (half scavenging capacity).

#### 4.6.2. The 2,2-Azino-Bis (3-Ethylbenzothiazoline-6-Sulfonic acid) Radical Scavenging Assay

The procedure was adapted from the methodology described by Thaipong et al., [32] and Arnao et al. [33] and fully developed for our research group in previous reported jobs [32] Briefly, a stable solution of ABTS* radical was prepared by reacting in darkness for 24 h, equal volumes of a 7.4 µM aqueous ABTS solution and 2.6 µM potassium peroxydisulfate. Working solution was prepared by adjusting an aliquot of the stable solution in methanol until an absorbance of 1.1 ± 0.02 was reached. An EPOCH microplate reader was used for absorbance reading at 734 nm and 96 microplates were used for the test. The same procedure as described for DPPH was followed for ABTS assay.

Half Scavenging capacity defined as the concentration of the EO required to scavenge 50% of the radical was calculated from the corresponding curve fitting of data (Radical scavenging percentage vs. EO concentrations). Radical scavenging percentages were calculated according to Equation (1):(1)$$\%\;RS=\left(1-\left(\frac{\mathrm{As}}{\mathrm{Ac}}\right)\right)\times 100$$
where,

Ac. Absorbance of the 100% radical adjusted solution.

As. Absorbance of the radical adjusted solution with the sample at different concentrations.

### 4.7. Anticholinesterase Assay

Acetylcholinesterase inhibitory effect of the essential oil was measured with a variation of the method described originally by Ellman et al. and reported in our previous jobs [28,33] and more recently in Andrade et al., [34]. Briefly, 20 µL of a 15 mM solution of acetylthiocoline in PBS pH 7.4 were placed along with 40 µL of Tris buffer pH 8.0 (containing 0.1 M NaCl and 0.02 M of MgCl_2_.6H_2_O), 100 µL of Ellman’s reagent solution in Tris buffer (DNTB, 3 mM) and 20 µL of EO sample solution. The reaction was placed in an EPOCH 2 microplate reader and preincubated for 3 min at 25 °C. Finally, the addition of 20 µL of the enzyme solution (0.5 U/mL) started the reaction. The amount of product released was monitored at 405 nm for 60 min. EO sample solutions were made by dissolving 10 mg of EO in MeOH. Three more 10 × factor dilutions and two intermediate log based doses were included to obtain final concentrations of 1000, 500, 100, 50 and 10 µg/mL and the experiments were performed by triplicate.

IC_50_ value was calculated from the corresponding curve fitting of data obtained from the calculated rate of reactions with Graph pad Prism 8.0.1 (San Diego, CA. USA) Donepezil hydrochloride was used as positive control.

## 5. Conclusions

The chemical composition and enantiomeric distribution of the EO of *H. strigosum,* as well as, a biological activity profile, were determined for the first time. This research contributes to our knowledge about the native aromatic species from Ecuador, particularly of the *Hedyosum* genus which is the only genus of the Chlorantaceae family, represented in Ecuador. The importance of aromatic species is well supported with this study where, an important antibacterial and antifungal effect was observed. Also, the antioxidant and anticholinesterase effect observed for this species encourage us to continue exploring our native flora, searching for novel candidates for the pharmaceutical or cosmetic industry.

## Figures and Tables

**Figure 1 plants-11-02832-f001:**
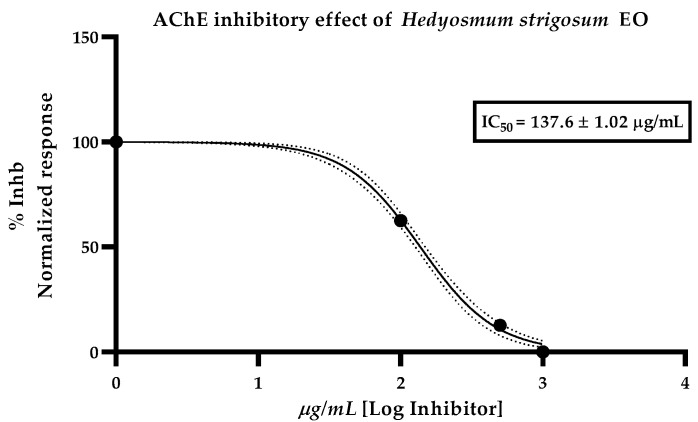
Inhibitory effect plot of *H*. *strigosum*, EO, against Acetylcholinesterase. Data represent the media of three replicas and three different experiments analyzed by non-linear regression method.

**Table 1 plants-11-02832-t001:** Chemical compounds present in the essential oil of the leaves of H. strigosum.

N°	Compounds	DB-5 ms			HP-INNOWax			
		LRI ^a^	LRI ^b^	%	LRI ^a^	LRI ^b^	%	CF
1	*α*-Thujene	923	924	0.17 ± 0.02	-	-	-	C_10_H_16_
2	*α*-Pinene	930	932	4.02 ± 0.02	1025	1028	3.92 ± 0.37	C_10_H_16_
3	Camphene	945	946	0.61 ± 0.01	1070	1075	0.51 ± 0.05	C_10_H_16_
4	Sabinene	968	969	0.95 ± 0.01	1113	1119	1.23 ± 0.14	C_10_H_16_
5	*β*-Pinene	973	974	1.43 ± 0.02	1125	1122	0.81 ± 0.09	C_10_H_16_
6	Myrcene	986	988	0.95 ± 0.04	-	-	-	C_10_H_16_
7	*α*-Phellandrene	1003	1002	12.15 ± 0.16	1168	1160	13.93 ± 1.09	C_10_H_16_
8	*α*-Terpinene	1013	1014	0.13 ± 0.01	1184	1186	0.21 ± 0.10	C_10_H_16_
9	*p-*Cymene <ρ->	1020	1020	3.88 ± 0.10	1277	1260	2.31 ± 0.35	C_10_H_16_
10	Limonene	1025	1024	3.70 ± 0.09	1204	1204	3.06 ± 0.39	C_10_H_16_
11	*β*-Phellandrene	1026	1025	2.03 ± 0.06	1212	1218	1.15 ± 0.32	C_10_H_16_
12	1,8-Cineole	1028	1026	4.7 ± 0.12	1215	1220	4.15 ± 0.45	C_10_H_18_O
13	(*Z*)-*β*-Ocimene	1032	1032	tr	1241	1233	0.58 ± 0.08	C_10_H_16_
14	(*E*)-*β*-Ocimene	1043	1044	0.32 ± 0.01	1258	1260	0.29 ± 0.04	C_10_H_16_
15	Terpinolene	1080	1086	0.11 ± 0.02	1286	1288	0.19 ± 0.02	C_10_H_16_
16	*p-*Cymenene	1086	1089	0.67 ± 0.02	1445	1438	0.33 ± 0.27	C_10_H_14_
17	Filifolone	-	-	-	1448	1445	0.55 ± 0.20	C_10_H_14_O
18	Linalool	1098	1095	8.73 ± 0.12	1571	1556	5.82 ± 0.19	C_10_H_18_O
19	Chrysanthenone	1117	1124	1.29 ± 0.13	1515	1489	5.02 ± 0.77	C_10_H_14_O
20	(*Z*)-*p-*Menth-2-en-1-ol	1121	1118	1.28 ± 0.02	1618	1638	0.41 ± 0.04	C_10_H_18_O
21	(*E*)-*p-*Menth-2-en-1-ol	1139	1136	0.47 ± 0.02	1587	1571	0.27 ± 0.05	C_10_H_18_O
22	Camphor	1142	1141	1.67 ± 0.04	1521	1522	1.85 ± 0.10	C_10_H_16_O
23	Pinocarvone	1157	1160	2.13 ± 0.05	1575	1580	1.19 ± 0.16	C_10_H_14_O
24	(*Z*)-Pinocamphone	1171	1172	0.40 ± 0.03	1550	1555	0.26 ± 0.09	C_10_H_16_O
25	(*Z*)-Piperitol	1191	1195	0.63 ± 0.03	1766	1758	0.34 ± 0.01	C_10_H_18_O
26	Thymol, methyl ether	1226	1232	1.50 ± 0.01	1603	1586	0.19 ± 0.03	C_11_H_16_O
27	Neral	1235	1235	0.69 ± 0.01	-	-	-	C_10_H_16_O
28	Geranial	1265	1264	0.62 ± 0.01	1749	1733	1.06 ± 0.08	C_10_H_16_O
29	Thymol	1289	1289	24.35 ± 0.27	2221	2189	22.48 ± 2.84	C_10_H_14_O
30	Thymol acetate	1341	1349	6.59 ± 0.23	1870	1840	9.39 ± 1.12	C_12_H_16_O_2_
31	Methyl eugenol	1396	1403	4.15 ± 0.04	2039	2023	0.81 ± 0.14	C_11_H_14_O_2_
32	*α*-Cedrene	1407	1410	0.23 ± 0.02	-	-	-	C_15_H_24_
33	*β*-Cedren	1410	1419	0.39 ± 0.01	-	-	-	C_15_H_24_
34	Germacrene D	1472	1480	3.17 ± 0.11	1700	1710	5.70 ± 0.44	C_15_H_24_
35	Bicyclogermacrene	1487	1500	0.84 ± 0.20	1725	1706	1.04 ± 0.11	C_15_H_24_
36	(*E*)-Methyl isoeugenol	1490	1491	0.04 ± 0.01	2196	2185	0.20 ± 0.02	C_11_H_14_O_2_
37	*δ*-Amorphene	1510	1511	0.32 ± 0.02	-	-	-	C_15_H_24_
38	Elemol	1542	1548	0.36 ± 0.12	2099	2090	0.67 ± 0.07	C_15_H_26_O
39	(*E)-*Nerolidol	1557	1561	0.88 ± 0.02	2063	2050	1.04 ± 0.11	C_15_H_26_O
40	Spathulenol	1568	1577	0.79 ± 0.04	2139	2140	0.32 ± 0.04	C_15_H_24_O
41	(*E*)-Isoeugenol acetate	1601	1614	0.39 ± 0.15	-	-	-	C_12_H_14_O_3_
42	*α-*Eudesmol	-	-	-	2235	2233	0.25 ± 0.08	C_15_H_26_O
43	*β*-Eudesmol	-	-	-	2243	2240	0.52 ± 0.15	C_15_H_26_O
44	*α-*Cadinol	-	-	-	2249	2225	0.22 ± 0.02	C_15_H_26_O
	ALM			31.13			28.52	
	OXM			46.96			43.40	
	HS			4.95			6.74	
	OS			2.03			3.02	
	Others			12.67			10.59	
	Total			97.74			92.27	

Percentage (%) is expressed as mean ± SD (standard deviation); LRI^a^: Calculated Linear Retention Index; LRI^b^: Linear Retention Index read in bibliography; tr: trace (<0.03). CF: Condensed formula. ALM. Aliphatic monoterpene hydrocarbons, OXM. Oxygenated monoterpenes, sesquiterpene hydrocarbons, Oxigenated sesquiterpenes.

**Table 2 plants-11-02832-t002:** Enantiomeric analysis from *H. strigosum* essential oil.

LRI ^a^	Component	Enantiomeric Distribution (%)	e.e. (%)
934	(*1R*,*5S*)-(–)-*α*-thujene	100	100
941	(*1S*,*5S*)-(–)-*α*-pinene	100	100
955	(*1R*,*4S*)-(–)-camphene	100	100
983	(*1R*,*5R*)-(+)-*β*-pinene	33.9	12.3
985	(*1S*,*5S*)-(–)-*β*-pinene	56.1	
989	(*1S*,*5S*)-(+)-sabinene	100	100
1006	(*R*)-(–)-*α*-phellandrene	33.7	32.7
1011	(*S*)-(+)-*α*-phellandrene	66.3	
1050	(*S*)-(–)-limonene	100	100
1194	(*R*)-(–)-linalool	44.1	11.9
1198	(S)-(+)-linalool	55.9	

LRI ^a^: Linear Calculated Retention Index; e.e: enantiomeric excess.

**Table 3 plants-11-02832-t003:** Minimum inhibitory concentration calculated for *H. strigosum* essential oil against ten human pathogenic microorganisms.

Microorganism	*H. strigosum* EO	Antimicrobial Agent
	MIC µg/mL	
Gram negative bacteria		
*Escherichia coli*	2000.00	1.56
*Pseudomonas aeruginosa*	NA	<0.39
*Salmonella enterica*	4000.00	<0.39
*Campylobacter jejuni*	125.00	
Gram positive bacteria		
*Listeria monocytogenes*	NA	1.56
*Enterococcus faecalis*	4000.00	0.78
*Enterococcus faecium*	2000.00	<0.39
*Staphylococcus aureus*	1000.00	<0.39
Fungi		
*Trychophyton rubrum*	125.00	<0.12
*Trichophyton interdigitale*	125.00	<0.12

NA. non active at the maximum dose tested of 4000 µg/mL

**Table 4 plants-11-02832-t004:** Half scavenging capacity of *H. strigosum,* EO.

EO	ABTS	DPPH
*Hedyosmum strigosum*	SC_50_ (µg/mL—µM *) *±* SD	
	25.53 ± 0.48	1313.73 ± 18.13
Trolox *	24.72 ±1.03	28.97 ± 1.24

* Half scavenging capacity of Trolox is expressed in micromolar units.

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
