# Peer review of "Chemical and Biological Activity Profiling of Hedyosmum strigosum Todzia Essential Oil, an Aromatic Native Shrub from Southern Ecuador"

_plants, 2022, doi:10.3390/plants11212832_

Round 1

Reviewer 1 Report

The manuscript entitled „ Chemical and biological activity profiling of Hedyosmum strigosum Todzia essential oil, an aromatic native shrub from south-3 ern Ecuador” is of great interest and scientific value.

However, I have a few comments and questions:

·         The vast majority of citations are from the 21st century, while the literature on anticholinesterase assay [37] dates back to 1961. Are there no more modern methods? Has the original method been modified in ongoing research?

·         9 out of 39 citations are self-citations by the authors of the publication?

·         Are the plants studied sufficiently widespread worldwide, or can they be grown in different climatic zones, to be of practical relevance?

·         The compounds found in the oil of the plant under study are known active substances (thymol, thymol acetate, 1,2-phellandrene, linalool). In the discussion of the results, it would be useful to compare the activity of the individual compounds with the obtained essential oil.

·         On what basis was the chromatographic column and programme conditions selected (own research or literature)?

·         I could not find the supplementary materials?

·         The research would be more complete if carried out on material collected over a period of years (e.g. 3 years)

·         Particularly relevant would be bioassay data for pure enantiomers of the biologically active compounds present in the extract under investigation.

In general, the manuscript is suitable for publication in Plants after minor corrections.

Author Response

Prof. Dr. Dilantha Fernando 

Editor-in-Chief

Department of Plant Science, University of Manitoba, Winnipeg, MB R3T 2N2, Canada

Dear Dr. Dilantha

All the suggestions made by reviewers were carefully checked and considered to improve our final manuscript. All the answers to every inquiry are listed velow in order of appearance. We appreciate every recommendation and we will be grateful to obtain a response to our manuscript to continue with the process. All the authors agreed with the final version of the manuscript.

Reply to reviewers:

Answers to reviewer 1

The manuscript entitled “Chemical and biological activity profiling of Hedyosmum strigosum Todzia essential oil, an aromatic native shrub from south-3 ern Ecuador” is of great interest and scientific value.

The authors do thank to the Reviewer 1 for the time invested in reviewing our manuscript. We really appreciate for recognizing our efforts and for all the comments and concerns raised and we have taken them in consideration, changing the manuscript accordingly

  1. The vast majority of citations are from the 21st century, while the literature on anticholinesterase assay [37] dates back to 1961. Are there no more modern methods? Has the original method been modified in ongoing research?

Reference 37 has been deleted from the manuscript. It was introduced only as a reference of the original work described by Ellman et al. The acetylcholinesterase assay method has been adapted to work in our lab conditions taking into account Ellman’s original work and Reel et al [reference  ] as reference method. The assay under the tested conditions with Ellman’s reagent is the gold standard for preliminary screenings of natural matrices. There are some variations that include the use of new substrates and revealing reagents similar to Ellman’s reagent but for preliminary studies it is considered unnecesary.

  1. 9 out of 39 citations are self-citations by the authors of the publication?

Some of the self citations of our previous jobs included in the manuscript have been deleted and, the more recent and relevant works remain.

  1. Are the plants studied sufficiently widespread worldwide, or can they be grown in different climatic zones, to be of practical relevance?

As stated in the Introduction section, the Hedyosmum genus is widely distributed from México to Brazil and specially in South América, across the andean mountains at different altitudes and they are well recognized by their pleasant smell, thus, the determination of the chemical composition and their biological profile is considerd for our group as relevant since the point of view that plants should be studied preliminarly to identify their comercial value before to proceed with applied studies, however, further studies should be conducted to identify the responsible compound of the aforementioned activity.

  1. The compounds found in the oil of the plant under study are known active substances (thymol, thymol acetate, 1,2-phellandrene, linalool). In the discussion of the results, it would be useful to compare the activity of the individual compounds with the obtained essential oil.

In the manuscript, information about important compounds such as thymol and linalool and minor compounds and their posible colaboration to the activy observed for the Essential oil of H. strigosum is included and is stated as follow:

“Regarding to the anticholinesterase effect, the H. strigosum EO exerted a moderate inhibitory potency with an IC50 value of 137.6 µg/mL, much higher than the reported effect of H. barisiliense EO which exhibited an inhibition percentage of 69.82 % at a dose of 1mg/mL. Many species of Hedyosmum genus have been explored for their antibacterial or antioxidant potential and their chemical profile but little or nothing is said about their inhibitory potential against acetylcholinesterase, however, information related to chemical entities isolated from the EO of this genus and their inhibitory potential can be found in literature, such as thymol which is the characteristic occurring compound in Thymus vulgaris but, it was found as the majority compound in our EO. As reported by Jukic et al [28], thymol presented an IC50 of 0.74 mg/mL and linalool was inactive at a dose higher than 1 mg/mL, surprisingly, carvacrol, the structural isomer of thymol, exerted a ten times higher potency with an IC50 of 0.063 mg/mL against acetylcholinesterase, suggesting that the hydroxyl functionality in -orto position, like occurring in carvacrol, plays an important role for the AChE inhibitory effect.

Despite its low concentration in the EO of H. strigosum, terpenes like terpinolene, terpinen-4-ol and E-caryophyllene (< to 1 %) can be collaborating for the moderate AChE inhibitory effect observed. As reported by Bonesi et al., terpinolene presented an IC50 of 156.4 µg/mL [29], meanwhile, terpinen-4-ol and E-caryophyllene exhibited a moderate effect with inhibition percentages of 21.4 % at 1.2mM and 32 % at 0.06mM doses, respectively. There is no report about the AchE inhibitory effect of α-phellandrene, the second major compound occurring for H. strigosum EO, however, in the same study of Bonesi, the AChE inhibitory effect of β-phelandrene is reported with an IC50 value of 120.2 µg/mL. highlighting the importance of terpenes containing EO and their AChE inhibitory capacity.”

  1. On what basis was the chromatographic column and programme conditions selected (own research or literature)?

Our research group has built a vast experience in the identification of chemical entities ocurring in essential oils through our well validated methods. First, the essential oil sample is subjected to preliminary runs, in which the parameters are varied until achieving a good separation and resolution of the chromatographic peaks. The starting parameters in these preliminary runs are chosen from experience and literature.

  1. I could not find the supplementary materials?

It is not mandatory for authors to submit supplementary material at Plants journal, however, all chromatographic and biological data analyzed are well supported in our manuscript and if any reviewer consider relevant to publish a supplementary material we will be able to do it under specific request.

  1. The research would be more complete if carried out on material collected over a period of years (e.g. 3 years)

It is well known that collecting plant samples in different phenological states or even in different climatological conditions will affect yield or chemical composition. Our pourpose as a research group was to identify, preliminarily, a plant as promising, by the identification of their chemical profile and its biological activity, report the findings and, later, if its interesting from different point of views, we will continue studying the potential applications of the essential oil obtained from it. Al lof it will depend of the yield and outcomes but for now, we consider that the information given is robust enough to continue studying the genus. We appreciate the suggestion made by the reviewer and we will consider such proposal for a new manuscript.

  1. Particularly relevant would be bioassay data for pure enantiomers of the biologically active compounds present in the extract under investigation.

Our research group is working in the identification of the biological profile of pure commercial compounds as Pinene (α and β) and other naturally ocurring terpenes but, it will be part of a different manuscript, however, a literature review of the main identifed compounds in the essential oil has been done in the manuscript by now. We appreciate the suggestion made by the reviewer and we will consider such a proposal for a new manuscript.

Thank you for your time and concern.

With best regards,

Luis Cartuche

Departamento de Química

Universidad Técnica Particular de Loja,

P.O. Box 11 01 608, Loja - Ecuador,

Tel: +593 7 370 1444,

E-mail address: [email protected]

Reviewer 2 Report

This manuscript describes the usefulness of essential oil obtained from Hedyosmun strigosum, which is a rare plant in central and south America area. Authors showed the chemical content of the essential oil by GC and its anti-microbial and anti-acetylcholinesterase activities. These presented data should be interesting to the readers of the journal. However, some experiments need some extra experiments in order to show the reliability of their presented data. Thus, this manuscript should be accepted after major revision. Here are some comments which should be referred for revision.

1, The expression of chemical formula in Table 1 should be corrected. The number should be in subscript.

2, In Table 2, the stereochemistries of the compounds should be written in italic. Moreover, the some lines in the table are missing.

3, In Table 3, authors should present MIC values for positive controls. Moreover, some numbers are too big.

4, In Table 4, all the numbers are written in italic. They should be written in normal type except for the scientific name of the microorganism.

5, In Figure 1, more data points were needed to calculate IC50 for reliability. Moreover, the numbers of experiments performed should be clarified.

6, All the scientific names of organisms should be written in italic.

7, The manuscript should receive a grammar check by a native speaker of English.

Author Response

Prof. Dr. Dilantha Fernando 

Editor-in-Chief

Department of Plant Science, University of Manitoba, Winnipeg, MB R3T 2N2, Canada

Dear Dr. Dilantha

All the suggestions made by reviewers were carefully checked and considered to improve our final manuscript. All the answers to every inquiry are listed velow in order of appearance. We appreciate every recommendation and we will be grateful to obtain a response to our manuscript to continue with the process. All the authors agreed with the final version of the manuscript.

Reply to reviewers:

Answers to reviewer 2

The authors do thank to the Reviewer 2 for the time invested in reviewing our manuscript. We appreciated all the corrections and comments raised and we have taken them into consideration, changing the manuscript accordingly.

  1. The expression of chemical formula in Table 1 should be corrected. The number should be in subscript.

We appreciate the suggestion.

Chemical formula (CF) in table 1 were corrected.

  1. In Table 2, the stereochemistries of the compounds should be written in italic. Moreover, the some lines in the table are missing.

Thank you.

All the suggestions made by reviewer have been corrected in Table 2

  1. In Table 3, authors should present MIC values for positive controls. Moreover, some numbers are too big.

Thank you. MIC values for antimicrobial agents were corrected to introduce only two decimal positions, as well as, MIC values for essential oil.

  1. In Table 4, all the numbers are written in italic. They should be written in normal type except for the scientific name of the microorganism.

Thank you. Typographic correction as suggested by reviewer was made in Table 4.

  1. In Figure 1, more data points were needed to calculate IC50 for reliability. Moreover, the numbers of experiments performed should be clarified.

A log base doses system was included to analyze acetylcholinesterase activity as mentioned in methodology:

“EO sample solutions were made by dissolving 10 mg of EO in MeOH. Three more 10 × factor dilutions and two intermediate log based doses were included to obtain final concentrations of 1000, 500, 100, 50 and 10 µg/mL and the experiments were performed by triplicate.

We considered of importance to built graph 1 based only including relevant data to represent the AChE inhibtory effect. Adittionally, the inclusion of a positive control as donepezil hydroclhoride validated the method as reviewed in literature.

  1. All the scientific names of organisms should be written in italic.

Thank you. Scientific names for microorganisms were carefully reviewed in the whole manuscript and corrected as suggested.

  1. The manuscript should receive a grammar check by a native speaker of English.

The whole manuscript has been carefully reviewed to avoid gramatical errors.

Thank you for your time and concern.

With best regards,

Luis Cartuche

Departamento de Química

Universidad Técnica Particular de Loja,

P.O. Box 11 01 608, Loja - Ecuador,

Tel: +593 7 370 1444,

E-mail address: [email protected]

Reviewer 3 Report

Dear, Luis Cartuche, Ph.D.

Please check and revise.

-------------------------------------------------------------------------------------------------------------------------

Table 1 and 2. You should describe E, Z, R, S, alpha, beta, etc. in Italic type. Generally, they are described in not Roman type but Italic type.

-------------------------------------------------------------------------------------------------------------------------

Author Response

Prof. Dr. Dilantha Fernando 

Editor-in-Chief

Department of Plant Science, University of Manitoba, Winnipeg, MB R3T 2N2, Canada

Dear Dr. Dilantha

All the suggestions made by reviewers were carefully checked and considered to improve our final manuscript. All the answers to every inquiry are listed velow in order of appearance. We appreciate every recommendation and we will be grateful to obtain a response to our manuscript to continue with the process. All the authors agreed with the final version of the manuscript.

Reply to reviewers:

Answers to reviewer 3

All the authors really appreciate the effort made by reviewer 3 to improve our manuscript.

  1. Table 1 and 2. You should describe E, Z, R, S, alpha, beta, etc. in Italic type. Generally, they are described in not Roman type but Italic type.

Thank you. All the suggestions made by the reviewer were considered for tables 1 and 2.

Thank you for your time and concern.

With best regards,

Luis Cartuche

Departamento de Química

Universidad Técnica Particular de Loja,

P.O. Box 11 01 608, Loja - Ecuador,

Tel: +593 7 370 1444,

E-mail address: [email protected]

Round 2

Reviewer 2 Report

This revised manuscript was corrected properly according to the reviewer's comment. Thus, this manuscript should be accepted as it is. However, authors are advised to check the English grammar carefully again for the publication.